# Antimicrobial Activity of Peptide-Coupled Antisense Peptide Nucleic Acids in *Streptococcus pneumoniae*

Gina Barkowsky,[a]* Corina Abt,[a] Irina Pöhner,[a]§ Adam Bieda,[a] Sven Hammerschmidt,[b] Anette Jacob,[c,d] [ID] Bernd Kreikemeyer,[a] [ID] Nadja Patenge[a]

[a]Institute of Medical Microbiology, Virology and Hygiene, University Medicine Rostock, Rostock, Germany

[b]Department of Molecular Genetics and Infection Biology, Interfaculty Institute for Genetics and Functional Genomics, Center for Functional Genomics of Microbes, University of Greifswald, Greifswald, Germany

[c]Peps4LS GmbH, Heidelberg, Germany

[d]Functional Genome Analysis, Deutsches Krebsforschungszentrum, Heidelberg, Germany

Gina Barkowski and Corina Abt contributed equally to this work. Author order was determined in order of decreasing seniority.

**ABSTRACT** *Streptococcus pneumoniae* is the most common cause of community-acquired pneumonia and is responsible for multiple other infectious diseases, such as meningitis and otitis media, in children. Resistance to penicillins, macrolides, and fluoroquinolones is increasing and, since the introduction of pneumococcal conjugate vaccines (PCVs), vaccine serotypes have been replaced by non-vaccine serotypes. Antisense peptide nucleic acids (PNAs) have been shown to reduce the growth of several pathogenic bacteria in various infection models. PNAs are frequently coupled to cell-penetrating peptides (CPPs) to improve spontaneous cellular PNA uptake. In this study, different CPPs were investigated for their capability to support translocation of antisense PNAs into *S. pneumoniae*. HIV-1 TAT- and $(RXR)_4$XB-coupled antisense PNAs efficiently reduced the viability of *S. pneumoniae* strains TIGR4 and D39 *in vitro*. Two essential genes, *gyrA* and *rpoB*, were used as targets for antisense PNAs. Overall, the antimicrobial activity of anti-*gyrA* PNAs was higher than that of anti-*rpoB* PNAs. Target gene transcription levels in *S. pneumoniae* were reduced following antisense PNA treatment. The effect of HIV-1 TAT- and $(RXR)_4$XB-anti-*gyrA* PNAs on pneumococcal survival was also studied *in vivo* using an insect infection model. Treatment increased the survival of infected *Galleria mellonella* larvae. Our results represent a proof of principle and may provide a basis for the development of efficient antisense molecules for treatment of *S. pneumoniae* infections.

**IMPORTANCE** *Streptococcus pneumoniae* is the most common cause of community-acquired pneumonia and is responsible for the deaths of up to 2 million children each year. Antibiotic resistance and strain replacement by non-vaccine serotypes are growing problems. For this reason, *S. pneumoniae* has been added to the WHO "global priority list" of antibiotic-resistant bacteria for which novel antimicrobials are most urgently needed. In this study, we investigated whether CPP-coupled antisense PNAs show antibacterial activity in *S. pneumoniae*. We demonstrated that HIV-1 TAT- and (RXR)4XB-coupled antisense PNAs were able to kill *S. pneumoniae in vitro*. The specificity of the antimicrobial effect was verified by reduced target gene transcription levels in *S. pneumoniae*. Moreover, CPP-antisense PNA treatment increased the survival rate of infected *Galleria mellonella* larvae *in vivo*. Based on these results, we believe that efficient antisense PNAs can be developed for the treatment of *S. pneumoniae* infections.

**KEYWORDS** antimicrobial activity, antimicrobial therapy, antisense molecules, pneumococcus, *Streptococcus pneumoniae*, peptide nucleic acid, cell penetrating peptide

Address correspondence to Nadja Patenge, Nadja.patenge@med.uni-rostock.de.

*Present address: Gina Barkowsky, Euroimmun, Dassow, Germany.

§Present address: Irina Pöhner, Institute for Anatomy, University Medicine Rostock, Rostock, Germany.

The authors declare no conflict of interest.

*S*treptococcus pneumoniae (pneumococcus) are Gram-positive pathogens which colonize the upper respiratory tract and cause lower respiratory tract infections, otitis media in children, conjunctivitis, bacteremia, and meningitis (1). Pneumococci are the main cause of community-acquired pneumonia (CAP), which mostly affects children under the age of five, the elderly, and patients with underlying comorbidities (2, 3). Infections occur more frequently during the cold season and in conjunction with viral infections of the upper respiratory tract (4, 5). The global burden of pneumococcal disease is high, being responsible for the deaths of up to 2 million children each year, most of whom live in developing countries (6). Pneumococcal pneumonia can be prevented by immunization and adequate nutrition, and by addressing environmental factors.

The introduction of pneumococcal conjugate vaccines (PCV) reduced invasive pneumococcal disease and carriage rates in immunized children, leading to protection of the unimmunized population, also referred to as herd immunity (7, 8). Globally, pneumococcal deaths in children declined by approximately 51% from 2000 to 2015 (9). Over the years, serotype replacement has resulted in the rising prevalence of non-vaccine serotypes in carriage and disease, limiting the protective effect of PCV (10). The genetic adaptability of *S. pneumoniae* facilitates the generation of recombinants which adjust to the selective pressures exerted by vaccines and antimicrobials, leading to the spread of antibiotic resistance and evasion of vaccine-induced immunity (11, 12). Consequently, *S. pneumoniae* is included in the WHO's "global priority list" (priority 3) of antibiotic-resistant bacteria for which novel antimicrobials are most urgently needed (13). To limit the use of broad-spectrum antibiotics, the development of species-specific antimicrobials is desirable (14). Antisense oligonucleotides (ASOs) targeting the start codon region of mRNAs have the potential to act specifically on the species level (15). Inhibiting the initiation of the translation of essential genes or virulence factors affects bacterial survival or pathogenesis, respectively.

Peptide nucleic acids (PNAs) are synthetic DNA analogues featuring a pseudopeptide backbone to which nucleobases are bound. PNAs can hybridize to DNA and RNA with high affinity (16). Their chemical composition, together with their resistance to proteases and nucleases, renders PNAs very stable (17). These properties make them interesting tools for gene silencing. However, surface barriers, including membranes, cell walls, and capsules, restrict bacterial uptake. Translocation of PNAs into cells can be facilitated by coupling them to cell-penetrating peptides (CPPs), which have been widely studied for cargo delivery into eukaryotic cells (18).

The efficiency of CPP-PNA uptake into bacteria depends on the species and can even vary between strains and serotypes depending on different surface properties. The antimicrobial effects of CPP-antisense PNAs have been intensively studied in Gram-negative bacteria (19). So far, a limited number of CPPs have identified which effectively support PNA uptake into Gram-positive bacteria, including *Listeria monocytogenes* and *Staphylococcus aureus* (20, 21). In a recent study, we systematically screened for CPPs which facilitate PNA translocation into *Streptococcus pyogenes*. We identified three CPPs which were able to support antimicrobial activity of anti-*gyrA* PNAs: HIV-1 TAT, K8, and (RXR)$_4$XB (22). Here, we investigated the effects of HIV-1 TAT, K8, and (RXR)$_4$XB on antisense PNA activity in *S. pneumoniae*.

## RESULTS

**Design of CPP-coupled antisense PNAs specific for *S. pneumoniae*.** In a previous study, we observed the antimicrobial effects of peptide-coupled anti-*gyrA* antisense PNAs specific for *S. pyogenes* (22). Because carrier molecules influence cargo uptake in a species-specific manner, we wanted to identify CPPs which mediated the uptake of antisense PNAs into *S. pneumoniae*. As antisense targets, we selected the essential genes *gyrA* and *rpoB*, which encode a subunit of the DNA gyrase and the bacterial RNA polymerase, respectively. These two enzymes have been used as antimicrobial targets in several studies, including antisense PNA experiments in Gram-positive pathogens (22–24). Peptides were coupled to PNAs via a flexible ethylene glycol linker (8-amino-3,

**TABLE 1** CPP-PNA conjugates for antisense-studies in *Streptococcus pneumoniae*[a]

| CPP | CPP-PNA designation | CPP-PNA sequence | Reference |
|---|---|---|---|
| HIV-1 TAT (48–57) | TAT-anti-*gyrA* PNA | GRKKRRQRRRYK-eg-tgcattaata | 47 |
| | TAT-anti-*gyrA* scPNA | GRKKRRQRRRYK-eg-aatgattact | |
| | TAT-anti-*rpoB* PNA | GRKKRRQRRRYK-eg-ctgccaagatga | |
| | TAT-anti-*rpoB* scPNA | GRKKRRQRRRYK-eg-cgtagtccaaag | |
| Oligolysin (K8) | K8-anti-*gyrA* PNA | KKKKKKKK-eg-tgcattaata | 31 |
| | K8-anti-*gyrA* scPNA | KKKKKKKK-eg-aatgattact | |
| (RXR)$_4$XB | (RXR)$_4$XB-anti-*gyrA* PNA | RXRRXRRXRRXRXB-eg-tgcattaata | 48 |
| | (RXR)$_4$XB-anti-*gyrA* scPNA | RXRRXRRXRRXRXB-eg-aatgattact | |
| | (RXR)$_4$XB-anti-*rpoB* PNA | RXRRXRRXRRXRXB-eg-ctgccaagatga | |
| | (RXR)$_4$XB-anti-*rpoB* scPNA | RXRRXRRXRRXRXB-eg-cgtagtccaaag | |

[a]CPP, cell-penetrating peptide; PNA, peptide nucleic acid; eg, 8-amino-3,6-dioxaoctan acid; X, 6-aminohexanoic acid; B, $\beta$-alanine.

6-dioxaoctanoic acid). The sequences of all peptide-conjugated PNAs used in this study are listed in Table 1. Scrambled control PNAs (scrambled PNAs, scPNAs) were composed of the same base pairs as the antisense PNAs but featured a randomized order.

**Antimicrobial effects of CPP-coupled antisense PNAs on *S. pneumoniae*.** The impact of different CPPs on the efficacy of antisense PNAs targeting *gyrA* and *rpoB* in *S. pneumoniae* was tested using an *in vitro* killing assay (25). *S. pneumoniae* strains were incubated for 6 h with CPP-antisense PNA conjugates. We determined the reduction of bacterial CFU/mL caused by different CPP-coupled antisense PNA constructs compared to an untreated control. Three different CPPs were tested: HIV-1 TAT (TAT), oligolysin (K8), and (RXR)$_4$XB (Table 1). Concentration-dependent bactericidal activity was investigated by treatment of *S. pneumoniae* TIGR4 in a CPP-antisense PNA conjugate concentration range of 2.5 to 20 $\mu$M (Fig. 1).

The CFU in treated samples was significantly reduced compared to the untreated control sample when TAT-anti-*gyrA* PNA was applied at a concentration range of 5 to 20 $\mu$M (Fig. 1A). A 50% inhibitory concentration (IC$_{50}$) of 0.4 $\mu$M was determined for TAT-anti-*gyrA* PNA. No reduction of bacterial counts was observed following incubation of *S. pneumoniae* TIGR4 with TAT-anti-*rpoB* PNA (Fig. 1B). TAT-conjugated scPNAs caused no significant reduction of CFU/mL following treatment (Fig. 1A/B), indicating a sequence-specific bactericidal effect of the TAT-anti-*gyrA* PNA construct. (RXR)$_4$XB-anti-*gyrA* PNA and (RXR)$_4$XB-anti-*rpoB* PNA caused significant decreases in *S. pneumoniae* TIGR4 bacterial counts following treatment (Fig. 1C/D).

In contrast, no reduction of CFU/mL was observed following incubation with K8-conjugated anti-*gyrA* PNA (data not shown). (RXR)$_4$XB-anti-*gyrA* was the most efficient construct, leading to a log CFU reduction of 4 following treatment with a concentration of 20 $\mu$M CPP-PNA. The IC$_{50}$ of (RXR)$_4$XB-anti-*gyrA* PNA was 0.63 $\mu$M. (RXR)$_4$XB-conjugated scPNAs caused a slight reduction in CFU/mL following treatment (log CFU reduction ≤1) (Fig. 1C and D). Considering the toxic effects of CPPs, we calculated the specific antimicrobial activity of the CPP-antisense PNAs by subtracting the log CFU reduction caused by CPP-scrambled control PNAs from the log CFU reduction caused by the corresponding CPP-antisense PNA (Table 2). In *S. pneumoniae* TIGR4, specific activities of log CFU reduction ≥1 were observed following treatment with 10 $\mu$M (RXR)$_4$XB-anti-*gyrA* PNA and TAT-anti-*gyrA* PNA, and with 20 $\mu$M (RXR)$_4$XB-anti-*rpoB* PNA. No specific activity could be detected using TAT-anti-*rpoB* PNA (Table 2).

Similar results were obtained with *S. pneumoniae* strain D39 (Fig. S1). (RXR)$_4$XB-coupled antisense PNAs were more efficient than TAT-coupled antisense PNAs. K8-antisense PNAs did not show any antimicrobial effect (data not shown). The difference between the target genes anti-*gyrA* and anti-*rpoB* was less pronounced in D39 than in TIGR4 (Table 2). No significant reduction of bacterial counts was observed following incubation with CPP-conjugated scPNAs (Fig. S1).

Compared to *S. pneumoniae* strains TIGR4 and D39, strain 19F was clearly affected

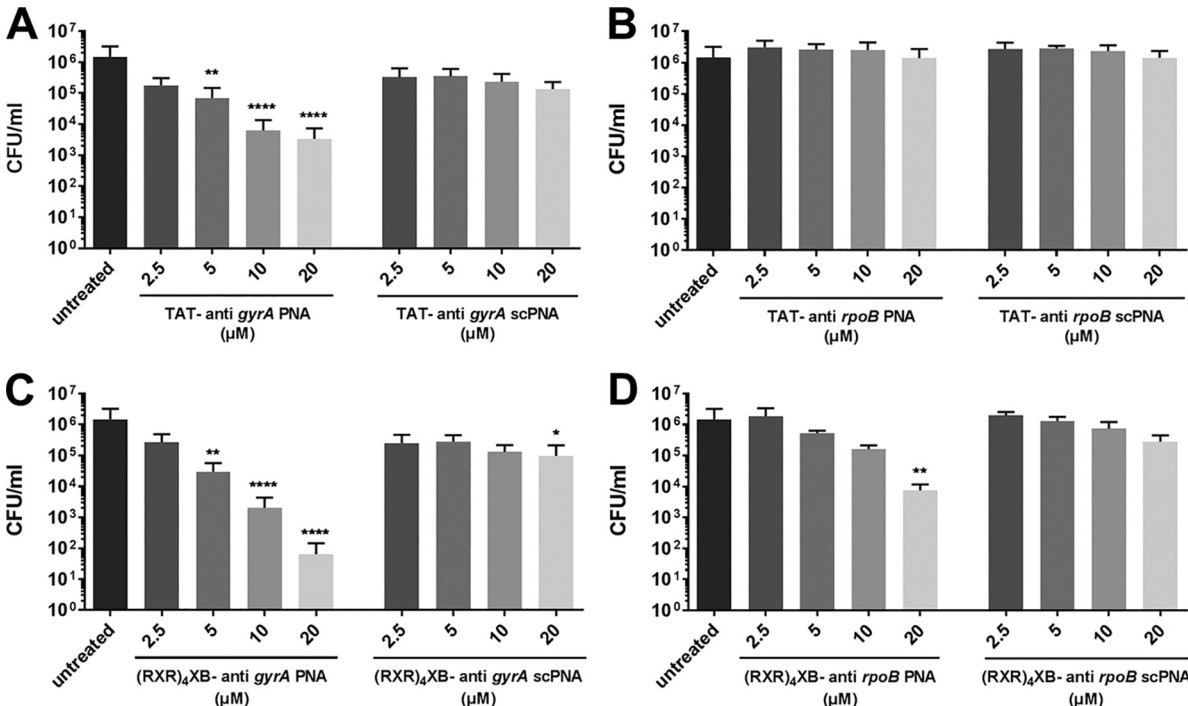

**FIG 1** Concentration-dependent reduction of the pneumococcal CFU/mL following treatment of *Streptococcus pneumoniae* TIGR4 with cell-penetrating peptide (CPP)-antisense peptide nucleic acids (PNAs) for 6 h. scPNA, scrambled PNA controls. (A) Treatment with TAT-anti-*gyrA* PNAs. (B) Treatment with TAT-anti-*rpoB* PNAs. (C) Treatment with (RXR)$_4$XB-anti-*gyrA* PNAs. (D) Treatment with (RXR)$_4$XB-anti-*rpoB* PNAs. Data are presented as means and standard deviation. Statistical significance was determined by one-way analysis of variance (ANOVA) with multiple comparisons. Differences between PNA conjugate samples and mock control (untreated) are shown as *, $P \leq 0.05$; **, $P \leq 0.01$; and ****, $P \leq 0.0001$. Sample size: $n = 5$.

by incubation with CPP-coupled scPNAs (Fig. S2). Specific antimicrobial activity of TAT-antisense PNAs was low, regardless of the target gene (Fig. S2, Table 2).

Treatment with (RXR)$_4$XB-anti-*gyrA* PNA caused a relevant reduction in *S. pneumoniae* strain 19F CFU. Treatment with 10 $\mu$M (RXR)$_4$XB-anti-*gyrA* PNA exhibited a specific log CFU reduction of $\geq$1 (Fig. S2C and D, Table 2). Overall, (RXR)$_4$XB-anti-*gyrA* PNA had the highest antimicrobial activity of all constructs tested. It achieved

**TABLE 2** Specific antimicrobial activity of CPP-antisense PNA conjugates in *Streptococcus pneumoniae*[a]

| CPP-PNA | C$_{PNA}$ ($\mu$M) | $\Delta$log CFU reduction[b] in *S. pneumoniae* strains | | |
|---|---|---|---|---|
| | | TIGR4 | D39 | 19F |
| (RXR)$_4$XB-anti-*gyrA* | 2.5 | 0 | 0 | 0.72 |
| | 5 | 0.98 | 0 | 1.20 |
| | 10 | 1.80 | 0.42 | 1.21 |
| | 20 | 3.18 | 2.23 | 1.48 |
| TAT-anti-*gyrA* | 2.5 | 0.27 | 0 | 0.50 |
| | 5 | 0.70 | 0 | 0.51 |
| | 10 | 1.57 | 1.30 | 0.22 |
| | 20 | 1.60 | 1.96 | 0.32 |
| (RXR)$_4$XB-anti-*rpoB* | 2.5 | 0 | 0 | 0.18 |
| | 5 | 0.40 | 0.16 | 0.17 |
| | 10 | 0.66 | 0.61 | 0.16 |
| | 20 | 1.58 | 1.98 | 0.56 |
| TAT-anti-*rpoB* | 2.5 | 0 | 0 | 0.05 |
| | 5 | 0 | 0.11 | 0.05 |
| | 10 | 0 | 0.16 | 0 |
| | 20 | 0 | 0.51 | 0 |

[a]CPP, cell-penetrating peptide; PNA, peptide nucleic acid.
[b]Log CFU reduction CPP-antisense PNA – log CFU reduction CPP-scPNA.

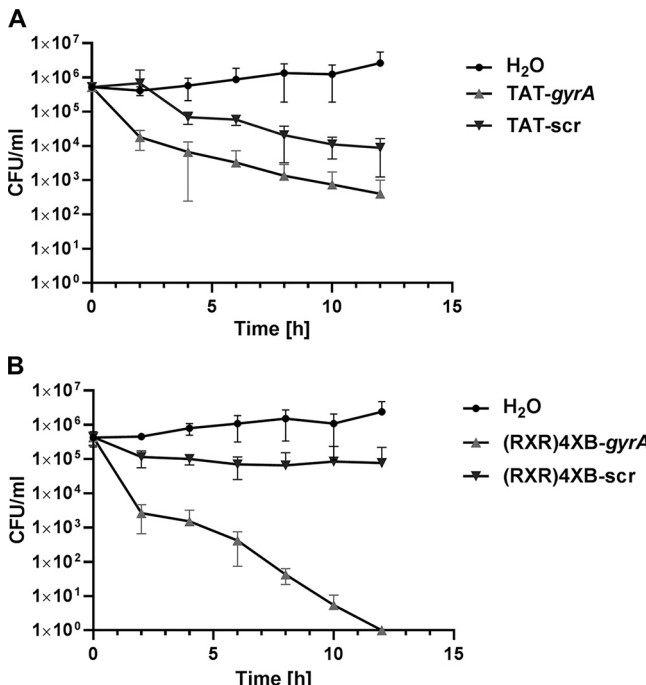

**FIG 2** Killing kinetics of CPP-anti-*gyrA* PNA treatment in *S. pneumoniae strain* TIGR4. (A) Bacterial counts following treatment with 10 $\mu$M TAT-anti-*gyrA* PNA or 10 $\mu$M TAT-anti-*gyrA* scPNAs. (B) Bacterial counts following treatment with 10 $\mu$M (RXR)$_4$XB-anti-*gyrA* PNAs or 10 $\mu$M (RXR)$_4$XB-anti-*gyrA* scPNAs. Data are presented as means and standard deviation. Sample size: $n = 3$.

a specific log CFU reduction of >1 in all *S. pneumoniae* strains investigated in this study (Table 2).

CPP-coupled scrambled PNAs were used as controls throughout this study because the cytotoxicity of CPPs is highly dependent on the cargo used (26). Accordingly, the antibacterial effects of (RXR)$_4$XB and TAT peptides on *S. pneumoniae* TIGR4 were higher than those of the conjugated peptides (Fig. S3).

**Bactericidal kinetics of CPP-coupled antisense PNAs in *S. pneumoniae*.** The antimicrobial activity kinetics of TAT- and (RXR)$_4$XB-conjugated anti-*gyrA* PNA constructs in *S. pneumoniae* TIGR4 were studied using an *in vitro* time-killing assay. Bacteria were treated with 10 $\mu$M CPP-antisense PNAs or scrambled control PNAs (Fig. 2). Samples were collected every 2 h following antisense treatment. CFU/mL was determined by plating of serial dilutions and plotted over time. Treatment with TAT-anti-*gyrA* PNA led to a steady reduction in bacterial counts, but no clearance was achieved over the course of the experiment (Fig. 2A). Following treatment with TAT-conjugated scPNA, reduced bacterial counts were observed 4 h post-treatment and continued throughout the experiment, indicating an unspecific toxic effect of the CPP. In contrast, (RXR)$_4$XB-conjugated anti-*gyrA* PNA treatment led to a continuous reduction in *S. pneumoniae* CFU until complete eradication after 12 h. (RXR)$_4$XB-conjugated scPNA did not cause reduced bacterial counts under these conditions.

**CPP-coupled antisense PNAs affect the abundance of target gene transcripts in *S. pneumoniae*.** We investigated the influence of treatment with CPP-conjugated antisense PNAs on the amount of target gene mRNA in *S. pneumoniae* using reverse transcription-quantitative PCR (RT-qPCR) (Fig. 3). *S. pneumoniae* strain TIGR4 was treated with a sublethal dose (log CFU reduction of <2) of CPP-PNA conjugates. Transcript abundance of the 5S rRNA gene was used for normalization (27). Target gene mRNA levels in mock-treated bacteria served as controls. Treatment with 7.5 $\mu$M CPP-anti-*gyrA* PNAs and 10 $\mu$M CPP-anti-*rpoB* PNAs led to significant reductions in the respective target gene transcripts compared to the untreated control sample (Fig. 3).

Following treatment with TAT-anti-*gyrA* PNA and (RXR)$_4$XB-anti-*gyrA* PNA, *gyrA* mRNA levels decreased to 29% and 45%, respectively (Fig. 3A). Transcript levels of *rpoB*

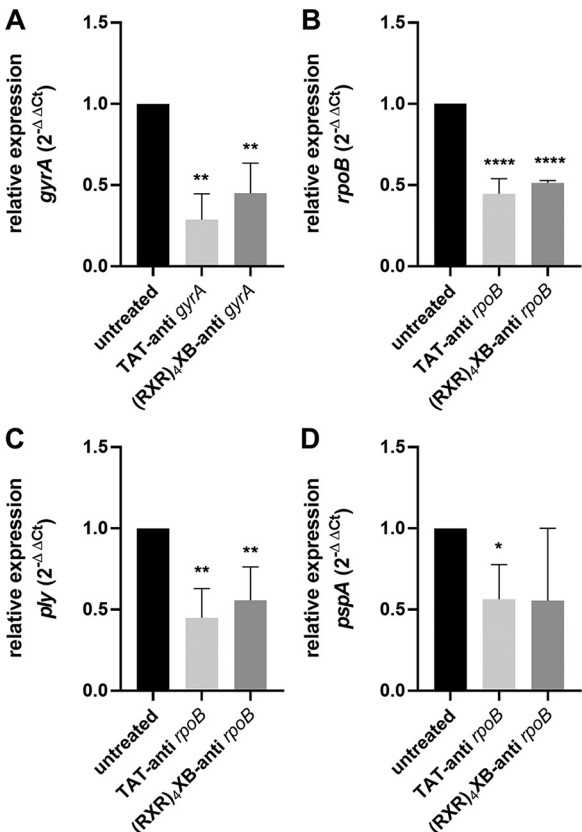

**FIG 3** Relative expression of target genes and virulence genes in *S. pneumoniae strain* TIGR4 following treatment with CPP-antisense PNAs. The 5S rRNA gene served as internal control. Relative expression was calculated using the threshold cycle ($2^{-\Delta\Delta CT}$) method. (A) Treatment with 7.5 $\mu$M TAT-anti-*gyrA* PNAs and 7.5 $\mu$M (RXR)$_4$XB-anti-*gyrA* PNAs. Relative expression of *gyrA*. (B) Treatment with 10 $\mu$M TAT-anti-*rpoB* PNAs and 10 $\mu$M (RXR)$_4$XB-anti-*rpoB* PNAs. Relative expression of *rpoB*. (C) Treatment with 10 $\mu$M TAT-anti-*rpoB* PNAs and 10 $\mu$M (RXR)$_4$XB-anti-*rpoB* PNAs. Relative expression of *ply*. (D) Treatment with 10 $\mu$M TAT-anti-*rpoB* PNAs and 10 $\mu$M (RXR)$_4$XB-anti-*rpoB* PNAs. Relative expression of *pspA*. Data are presented as means and standard deviation. Statistical significance was determined by one-way ANOVA with multiple comparisons. Differences between PNA conjugate samples and the mock control (untreated) are shown as *, $P \leq 0.05$; **, $P \leq 0.01$; and ****, $P \leq 0.0001$. Sample size: $n = 3$.

were reduced to 45% and 51%, respectively, upon incubation with TAT-anti-*rpoB* PNA and (RXR)$_4$XB-anti-*rpoB* PNA (Fig. 3B). The addition of higher concentrations of anti-*rpoB* PNAs was required to obtain a significant reduction in relative target gene expression. These results underline the higher antimicrobial efficiency of anti-*gyrA* PNAs we observed in *S. pneumoniae* TIGR4 (Fig. 1).

**Antisense PNA-mediated *rpoB* downregulation affects transcription of virulence genes.** Because the product of *rpoB* is the $\beta$-subunit of the bacterial RNA-polymerase, which is responsible for RNA synthesis, downregulation of *rpoB* by treatment with antisense PNAs is expected to affect gene expression. Therefore, we tested the relative expression of two virulence factor genes following treatment with 10 $\mu$M CPP-anti-*rpoB* PNAs compared to expression in the mock-treated control as an example (Fig. 3C and D).

Following TAT-anti-*rpoB* PNA treatment, the expression of *ply*, which codes for pneumolysin, was reduced to 45%. Treatment with (RXR)$_4$XB-anti-*rpoB* PNA reduced *ply* expression to 56% (Fig. 3C). Following treatment with TAT-anti-*rpoB* PNA, the expression of *pspA*, which codes for pneumococcal surface protein A, was reduced to 57%. In contrast, there was high variability in relative *pspA* transcription levels following treatment with (RXR)$_4$XB-anti-*rpoB* PNA, and no significant reduction could be observed (Fig. 3D).

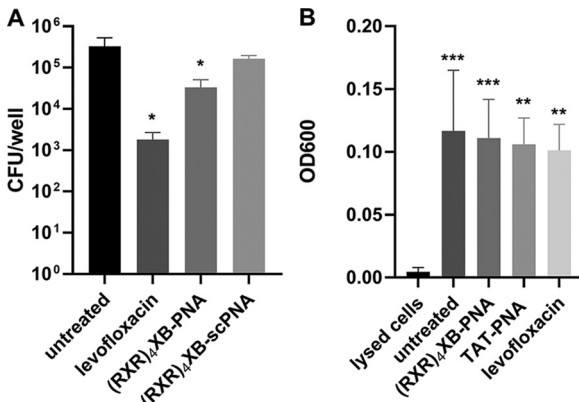

**FIG 4** Antimicrobial and cytotoxic effects of (RXR)4XB-anti-*gyrA* PNA in a cell-based infection system. (A) Detroit 562 cells infected with *S. pneumoniae* TIGR4 and subsequently treated with 1 $\mu$g levofloxacin, 20 $\mu$M (RXR)$_4$XB-anti-*gyrA* PNA, or 20 $\mu$M (RXR)$_4$XB-anti-*gyrA* scPNA, respectively. Sample size: $n$ = 3. Statistical significance was determined by one-way ANOVA with multiple comparisons. Differences between PNA conjugate samples and mock control (untreated) are shown as *, $P \leq 0.05$. (B) Detroit 562 cells were lysed and treated with 20 $\mu$M (RXR)$_4$XB-PNA or 20 $\mu$M TAT-PNA, respectively. Biomass was stained with crystal violet, bound crystal violet was extracted, and the optical density of the solution was measured at 600 nm. Sample size: $n$ = 4. Statistical significance was determined by one-way ANOVA with multiple comparisons. Differences between samples and maximal cell lysis are shown as **, $P \leq 0.01$; ***, $P \leq 0.001$.

**Antimicrobial effects of CPP-antisense PNA conjugates in a cell-based infection model.** Detroit 562 human pharyngeal epithelial cells were infected with *S. pneumoniae* TIGR4 at a multiplicity of infection (MOI) of 5. Cells were treated with 20 $\mu$M (RXR)$_4$XB-anti-*gyrA* PNA or scrambled control PNA, respectively. As a positive control, 1 $\mu$g levofloxacin, corresponding to 3$\times$ MIC, was added. Following incubation for 3 h at 37°C, bacterial CFU counts were determined for CPP-antisense PNA-treated samples and compared to those in nontreated samples and in samples containing scrambled control PNAs (Fig. 4A). Treatment with (RXR)$_4$XB-anti-*gyrA* caused a log CFU reduction of 1. Addition of levofloxacin led to a log CFU reduction of 2. No significant changes in bacterial CFU were observed following application of (RXR)$_4$XB-conjugated scPNAs in comparison to the untreated control.

We investigated the potential cytotoxic effects of CPP-PNAs on Detroit 562 cells by measuring cellular viability using a crystal violet assay (Fig. 4B). Cells were incubated with 20 $\mu$M (RXR)$_4$XB-PNA or TAT-PNA, respectively. Untreated cells or cells treated with levofloxacin served as negative controls. Cell lysis was used as positive control for cell death. No cell death was observed following treatment with levofloxacin or CPP-PNAs. Similar results were observed using a lactate dehydrogenase (LDH) leakage assay. LDH release after treatment of Detroit 562 cells was compared to maximum LDH release (Fig. S4). Application of 20 $\mu$M (RXR)$_4$XB-PNA, 20 $\mu$M TAT-PNA, or 1 $\mu$g levofloxacin caused 5%, 6%, or 3% of maximum LDH release, respectively. Together, these results indicate that CPP-PNA conjugates have low cytotoxic activity comparable to that of the conventional antibiotic levofloxacin.

**Evaluation of CPP-antisense PNA conjugates in a *G. mellonella* infection model.** Antimicrobial efficiency of CPP-antisense PNA conjugates was evaluated *in vivo* employing a *G. mellonella* infection and treatment model (22, 28). Larvae were infected with *S. pneumoniae* strain TIGR4 and subsequently treated with 10 nmol CPP-anti-*gyrA* PNAs. Treatment with levofloxacin served as a positive control. Larval survival was observed over 7 days.

The administration of 10 or 1 $\mu$g levofloxacin, corresponding to approximately 25$\times$ or 2.5$\times$ MIC, respectively, increased the survival of infected larvae from 19% to 74% or 40%, respectively (Fig. 5A). Larvae treated with TAT-anti-*gyrA* PNA or (RXR)$_4$XB-anti-*gyrA* PNA exhibited increased survival compared to mock-treated larvae or those treated with CPP-scPNA (Fig. 5B/C). The survival rate was 25% or 22% higher following

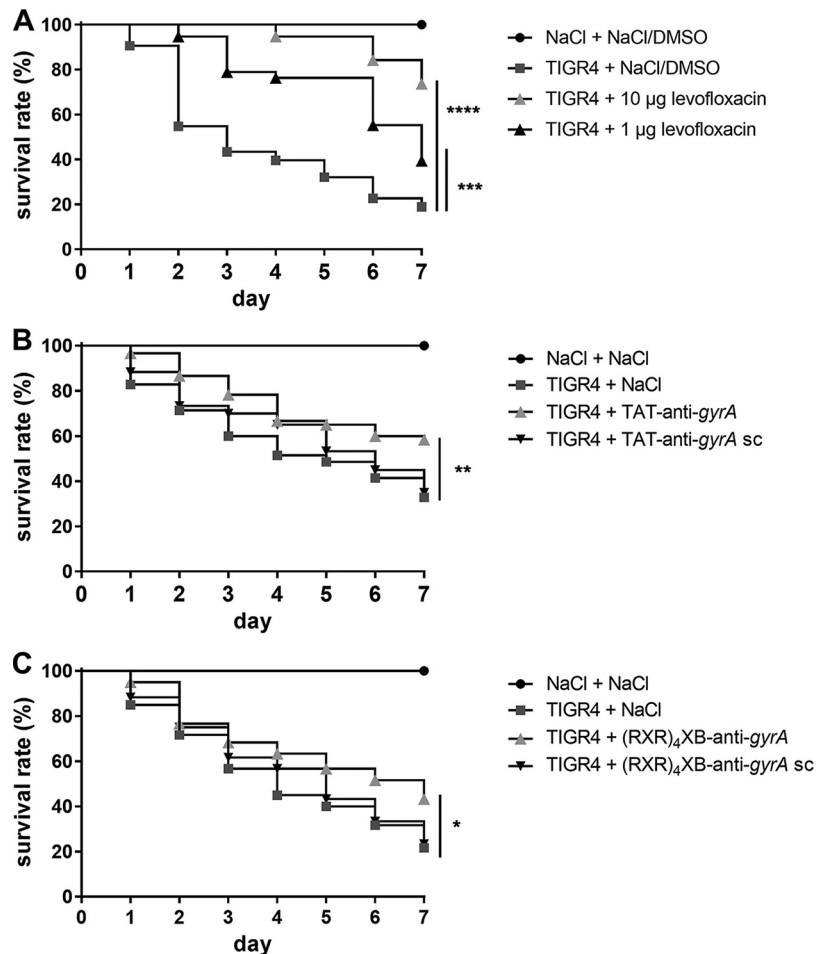

**FIG 5** Survival of *Galleria mellonella* larvae treated with 10 nmol CPP-PNAs following infection with *S. pneumoniae* TIGR4. (A) *G. mellonella* larvae infected with *S. pneumoniae* TIGR4 and subsequently treated with 10 or 1 μg levofloxacin. Sample size: *n* = 40 larvae per group. (B) *G. mellonella* larvae infected with *S. pneumoniae* TIGR4 and subsequently treated with TAT-anti-*gyrA* PNA or TAT-anti-*gyrA* scPNA, respectively. Sample size: *n* = 60 larvae per group. (C) *G. mellonella* larvae infected with *S. pneumoniae* TIGR4 and subsequently treated with (RXR)₄XB-anti-*gyrA* PNA or (RXR)₄XB-anti-*gyrA* scPNA, respectively. Sample size: *n* = 60 larvae per group. Statistical significance was determined using the log-rank test. Differences between curves are shown as *, $P \leq 0.05$; **, $P \leq 0.01$; ***, $P \leq 0.001$; and ****, $P \leq 0.0001$.

injection of TAT-anti-*gyrA* PNA or (RXR)₄XB-anti-*gyrA* PNA, respectively, which is comparable to the effect of 1 μg levofloxacin (21%). Treatment of *S. pneumoniae* strain TIGR4-infected larvae with TAT-anti-*rpoB* PNA or (RXR)₄XB-anti-*rpoB* PNA did not increase survival under these conditions (data not shown). These results are in accordance with the lower antimicrobial activity of these constructs compared to that of the anti-*gyrA* molecules we observed *in vitro* (Fig. 1).

To establish the antibacterial effect of PNAs in the invertebrate infection model, we assessed the bacterial loads in infected larvae (Fig. 6). Larvae were infected and subsequently treated with NaCl (mock), 10 nmol CPP-PNA, or levofloxacin (positive control), respectively. From each group, four larvae were homogenized at 24 h postinfection. CFU/larva were determined following serial dilutions of the suspension and plating on selective medium. Treatment with (RXR)₄XB-anti-*gyrA* caused a log CFU reduction of 1 compared to that in the mock-treated larvae. The addition of 1 or 10 μg levofloxacin led to a log CFU reduction of 2 or 6, respectively. In contrast, treatment with (RXR)₄XB-conjugated scPNAs caused no significant changes in the bacterial CFU in comparison to the untreated control, suggesting a target gene-specific antimicrobial effect in the infected larvae.

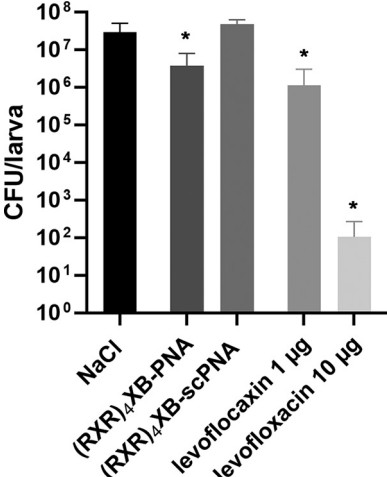

**FIG 6** Bacterial load of *G. mellonella* larvae following infection with *S. pneumoniae* TIGR4. *G. mellonella* larvae infected with *S. pneumoniae* TIGR4 and subsequently treated with either levofloxacin or (RXR)$_4$XB-PNAs. After 24 h, larvae were homogenized. Bacterial load was determined by serial dilution, plating on selective medium, and CFU calculation. Sample size: $n$ = 4 larvae per group in three independent experiments. Data are presented as means and standard deviation. Statistical significance was determined by one-way ANOVA with multiple comparisons. Differences between treated samples and mock control (NaCl) are shown as *, $P \leq 0.05$.

## DISCUSSION

Antisense molecules are potential alternative anti-infective therapeutics which provide several advantages compared to conventional antibiotics. They combine a high specificity toward their target structures with broad application versatility. Furthermore, there are no preexisting natural resistance mechanisms. It is expected that upon application of antisense therapeutics, genetic resistance will be developed. In this case, it is easy to respond in a timely manner by adjusting the respective antisense sequence. Nevertheless, several challenges remain before effective antisense agents can be identified. One major limitation in the use of antisense PNAs is their restricted uptake into cells. To improve translocation, many cargos, including PNAs, have been coupled to CPPs (29, 30).

In this work, we compared the antimicrobial efficiency of K8-, TAT-, and (RXR)$_4$XB-conjugated antisense PNAs in the *S. pneumoniae* strains TIGR4, D39, and 19F. K8-coupled antisense PNAs did not show bactericidal activity in any of the *S. pneumoniae* strains tested. Previously, we showed that K8-coupled anti-*gyrA* PNAs affected the growth of *S. pyogenes*, whereas no effect was mediated by polyarginine-coupled PNAs (22). In eukaryotic cells, polyarginine peptides were more efficient carriers than polylysines (31). R9 coupled to antimicrobial peptides was able to enhance bactericidal activity in Gram-negative bacteria (32). Arginine-rich CPPs use a passive translocation mechanism by inducing membrane multilamellarity and fusion (33). In future experiments, polyarginine should be studied in *S. pneumoniae* because its surface composition differs from that of *S. pyogenes* and might support the potency of polyarginine as a CPP.

(RXR)$_4$XB-conjugated antisense PNAs reduced bacterial counts of *S. pneumoniae* TIGR4, D39, and 19F in a killing assay. In all three strains, TAT-antisense PNA constructs were less efficient than (RXR)$_4$XB-coupled antisense PNAs. In a previous study, we observed a similar difference between these two CPPs in *S. pyogenes* (22). Accordingly, (RXR)$_4$XB-coupled anti-*rpoA* PNA caused the greatest reduction in CFU/mL in *L. monocytogenes*, followed by TAT-anti-*rpoA* PNA (20). Overall, (RXR)$_4$XB seems to be a potent CPP in Gram-positive bacteria.

CPP-mediated toxicity might challenge the application of carrier peptide-coupled PNAs as antimicrobial agents. In this work, we did not observe (RXR)$_4$XB-derived antibacterial effects in *S. pneumoniae* strains TIGR4 and D39. In contrast, TAT-scPNA reduced the number of *S. pneumoniae* strain TIGR4 slightly over the course of the time-killing experiment. At high concentrations, both TAT-scPNA and (RXR)$_4$XB-scPNA reduced CFU of *S. pneumoniae*

19F in the killing assay. Previously, we observed a slight nonspecific toxic effect of TAT in *S. pyogenes* at moderate concentrations (22). Together, these data highlight the fact that antimicrobial CPP-antisense PNA efficiency and specificity depend not only on the species but also on the respective strain investigated. *S. pneumoniae* serotypes are known to express biochemically distinct capsular polysaccharides (34). Capsules are among the bacterial surface barriers which limit CPP-PNA uptake. We hypothesize, therefore, that differences in capsule size and polysaccharide composition may underlay the differential susceptibility toward CPP-PNAs we observed in this study. The development of beneficial CPP-antisense PNA-based therapies requires extended screening for CPP-PNAs which exhibit antibacterial activity on the most prevalent clinical strains.

Typically, essential genes are used as antisense targets for antibacterial agents (35). We compared the efficiency of two different target genes, *gyrA* and *rpoB*. Both genes have been studied as antisense targets in bacteria and encode structures that are recognized by antibiotics (19, 23, 36). In *S. pneumoniae*, inhibition of *gyrA* expression was more efficient than targeting of *rpoB*. (RXR)$_4$XB-anti-*gyrA* PNA had the highest antibacterial efficiency in *S. pneumoniae* strains TIGR4, D39, and 19F. TAT-anti-*rpoB* PNA did not affect *S. pneumoniae* TIGR4 and D39. Only a slight antibacterial effect of TAT-anti *rpoB* PNA could be observed in *S. pneumoniae* 19F. (RXR)$_4$XB coupled to anti-*rpoB* PNA showed bactericidal activity comparable to that of (RXR)$_4$XB-anti-*gyrA* PNAs in strains D39 and 19F. We examined the target gene specificity of CPP-PNA treatment in *S. pneumoniae* TIGR4 by monitoring target gene transcript abundance in the presence of sublethal CPP-PNA concentrations. Under these conditions, moderate target gene mRNA reduction was observed regardless of the CPP-antisense PNA construct used. In conclusion, all four constructs were taken up by the bacterial cell and were able to bind to their respective targets. We hypothesize that distinct translocation efficacy, binding affinity, and other, as-yet unknown factors are responsible for the differences in dose-dependency and kill-kinetics between CPP-antisense PNAs present at lethal concentrations.

In *L. monocytogenes* and *S. aureus*, downregulation of *rpoA* translation caused reduced mRNA synthesis and thereby affected downstream genes, including virulence factor genes (20, 23). Similarly, we observed reduced expression of *ply* and *pspA* following treatment with anti-*rpoB* PNAs in *S. pneumoniae* strain TIGR4. Expression of *ply* was significantly reduced when TIGR4 was incubated with 10 $\mu$M TAT-anti-*rpoB* PNA, even though no bactericidal effect of this construct could be observed under these conditions. Knockdown of *rpoB* by antisense PNAs at concentrations which do not result in bacterial clearance will cause reduced virulence factor gene expression, thereby interfering with *S. pneumoniae* virulence and transmission. Some possible consequences of reduced *ply* expression by *S. pneumoniae* are decreased inflammation, shedding, and transmission (37).

In this study, (RXR)$_4$XB-anti-*rpoB* PNA caused downregulation of virulence factor genes. In a *G. mellonella* infection model, CPP-anti-*rpoB* PNAs did not increase survival under the conditions tested in this study. However, effects on shedding and transmission are not accounted for in this model. Compared to those in *S. pyogenes* and *L. monocytogenes*, the effective concentrations of TAT- and (RXR)$_4$XB-antisense PNAs were rather high in *S. pneumoniae* (20, 22). For instance, MIC values for all constructs were $\geq$62.5 $\mu$M (data not shown) compared to 15.6 $\mu$M for TAT-anti-*gyrA* PNA in *S. pyogenes* (22). Another strategy to improve antibacterial effects of antisense PNAs is a combination therapy with conventional antibiotics. Previous studies have demonstrated that synergistic effects support the bactericidal activity of peptide-coupled antisense PNAs in bacteria (22, 36).

In future studies, more potential targets may be investigated. In uropathogenic *Escherichia coli*, 11 essential genes with varying expression levels were investigated as potential targets for peptide-conjugated PNAs (38). Three promising target mRNAs were identified for effective growth inhibition: *dnaB*, *ftsZ*, and *rpsH*. This analysis also showed that transcript abundance does not predict target vulnerability. Furthermore,

the authors demonstrated that 9mer PNAs were generally as effective in inhibiting bacterial growth as their 10mer counterparts (38).

In addition to knockdown of essential genes, knockdown of virulence genes is becoming increasingly popular (39). Indirect knockdown by limited RNA polymerase production, but also direct antisense knockdown of *ply*, *pspA*, or other well-characterized virulence factors of *S. pneumoniae*, are potential measures to reduce infection-triggered damage. To tackle the rising number of multidrug-resistant invasive pneumococci, antisense targeting of resistance genes could be utilized to suppress bacterial mechanisms of antibiotic resistance. Numerous studies, in several Gram-negative organisms and in *S. aureus*, have used anti-resistance gene agents to increase antibiotic efficacy (40–43). In a similar approach, resistance genes could be targeted in *S. pneumoniae* to achieve resensitization of resistant strains (40, 41, 43).

Overall, we demonstrated that CPP-coupled antisense PNAs could affect the viability of three relevant *S. pneumoniae* serotypes. The effect was target gene-specific and could be verified *in vivo* using an insect infection model. In future studies, more potential carrier molecules and more target genes should be studied with the aim of identifying constructs with efficient antimicrobial activity toward *S. pneumoniae*. Promising peptide-coupled antisense PNAs could then be further investigated in murine infection models.

## MATERIALS AND METHODS

**PNA synthesis.** CPP-PNAs were synthesized and purified by high-pressure liquid chromatography (Peps4LS GmbH, Heidelberg, Germany) (44). The sequences of all CPP-PNAs used in this study are listed in Table 1.

**Bacterial strains and culture conditions.** *S. pneumoniae* strains were grown on Columbia blood agar plates (Becton Dickinson GmbH, Heidelberg, Germany) and cultivated to deceleration growth phase for a maximum of 10 h in brain heart infusion broth (BHI; Oxoid, Wesel, Germany) at 37°C under a 5% $CO_2$ atmosphere. Viability was verified by CFU/mL determination. We used serotype 2 strain D39 (NCTC 7466), serotype 19F (RKI 704), and serotype 4 strain TIGR4 (45).

**Bacterial killing assay.** After culturing *S. pneumoniae* strains in BHI, D39 and 19F were diluted to approximately $10^5$ CFU/mL in phosphate-buffered saline (PBS):BHI (93%:7%), while TIGR4 was diluted to approximately $10^5$ CFU/mL in PBS:BHI (96%:4%). Next, 450-$\mu$L volumes of the respective bacterial suspensions, containing approximately $0.5 \times 10^5$ CFU, were transferred to 2-mL reaction tubes. A 50-$\mu$L volume of PNA solution was added to a final PNA concentration of 1 to 20 $\mu$M or as indicated. Fifty $\mu$L $H_2O$ served as a mock control. The reaction tubes were incubated for 6 h at 37°C and 7 rpm (Rotor SB3; Stuart, Staffordshire, United Kingdom). The concentration of CPP-PNA that reduced CFU/mL by 50% after 6 h of incubation compared to an untreated control ($IC_{50}$) was calculated by nonlinear regression analysis using GraphPad Prism software (GraphPad Software, San Diego, CA).

For monitoring of killing kinetics, samples were collected 2 to 12 h post-treatment. Viable cell counts were determined by plating appropriate dilutions on BHI agar plates. CFU were determined by visual inspection following overnight incubation at 37°C under a 5% $CO_2$ atmosphere. At time point 0, the viable cell count corresponded to $1 \times 10^5$ to $3 \times 10^5$ CFU/mL. Experiments were performed in at least three independent biological replicates, as indicated in the figure legends.

**Extraction of total RNA.** For RNA isolation, 450 $\mu$L bacterial suspension ($10^5$ CFU/mL in BHI) was prepared for each experimental condition, treated with CPP-PNA conjugates as indicated, and incubated in a 2-mL reaction tube for 6 h at 37°C and 7 rpm (Rotor SB3; Stuart, Staffordshire, United Kingdom). Subsequently, five samples per condition were pooled. Bacteria were pelleted immediately, shock-frozen in liquid nitrogen, and stored at −80°C until use. Total RNA was extracted according to the protocol published by Li-Korotky et al. (46). Briefly, 30 $\mu$g polyinosinic acid (Poly I) was added to each bacterial pellet. TRIzol reagent was added to the bacteria/Poly I mix, which was subsequently disrupted in a homogenizer (Peqlab Biotechnologie GmbH, Erlangen, Germany). RNA was extracted twice with 200 $\mu$L chloroform (Thermo Fisher Scientific, Darmstadt, Germany). Following the addition of 5 $\mu$L glycogen and 500 $\mu$L isopropanol, RNA was precipitated at 13,000 rcf (relative centrifugal force) at 4°C for 15 min. Pellets were suspended in $H_2O$. RNA was subsequently treated with acid phenol:chloroform:isoamyl alcohol (125:24:1) (pH 4.5) (Thermo Fisher Scientific) and TURBO DNase (Thermo Fisher Scientific) according to the manufacturer's instructions. RNA was stored at −80°C until further use.

**Reverse transcription-quantitative PCR.** cDNA synthesis was performed using the Superscript First-Strand Synthesis system for RT-qPCR (Invitrogen, Thermo Fisher Scientific, Darmstadt, Germany). Quantitative PCR amplification was conducted with SYBR Green (Thermo Fisher Scientific) using the LightCycler 480 real-time PCR system (Roche Diagnostics GmbH, Mannheim, Germany). The 5S rRNA gene served as internal control. Relative expression was calculated using the $2^{-\Delta\Delta CT}$ method (27). Primers were designed based on the full genome sequence of *S. pneumoniae* strain TIGR4 (NCBI accession no. NZ_AKVY0100000). Primers are listed as follows: *gyrA*-specific, 5′-CTGAGTATGACCTCTTGGC-3′ and 5′-CCAATCAACTCTGTACGGC-3′; *rpoB*-specific, 5′-GGACCATACTCAACTGTTACCC-3′ and 5′-TCCATTCCTTCATCCAAGTCGC-3′; *ply*-specific, 5′-TTACGCACTAGTGGCAAATCGG-3′ and 5′-CTTTACAGCAGATCATCCAGGC-3′; *pspA*-specific, 5′-TTCCTGAACCA

AATGCGTTGGC-3′ and 5′-GTCTTACCTTCAGGATCAAGGC-3′; 5S-specific, 5′-CGATAGCCTAGGAGATACACC-3′ and 5′-GGGCTTAACTTCTGTGTTCGG-3′.

**Cell based infection model.** The human pharynx carcinoma epithelial cell line Detroit 562 (ATCC CCL138) was cultivated in Dulbecco's modified Eagle medium (DMEM)/F12 (1:1) (Thermo Fisher Scientific, Schwerte, Germany) supplemented with 10% fetal bovine serum at 37°C and 5% $CO_2$. Confluent Detroit 562 cells were infected with *S. pneumoniae* TIGR4 at an MOI of 5. Cultures were treated with 20 $\mu$M CPP-PNAs or 1 $\mu$g levofloxacin, respectively, and incubated for 3 h at 37°C, 5% $CO_2$. After incubation, the supernatant was collected. Adherent cells were detached with Accutase solution (Merck KGaA, Darmstadt, Germany), collected by centrifugation for 10 min at 4,000 × *g*, and combined with the supernatant. Cells were pelleted by centrifugation, washed twice with PBS, and suspended in PBS. CFU/well was determined following serial dilution and plating on Todd Hewitt Broth, supplemented with Yeast, (THY) agar plates.

**Cytotoxicity assays.** Confluent Detroit 562 cells were treated with 20 $\mu$M CPP-PNAs or 1 $\mu$g levofloxacin, respectively, and incubated for 3 h at 37°C, 5% $CO_2$. Crystal violet staining was performed to determine biomass. Cells were washed with PBS and fixed with 100 $\mu$L methanol for 10 min. Subsequently, cells were washed with PBS containing 0.005% Tween 20, incubated with 0.1% crystal violet solution for 10 min, and washed with $H_2O$. The remaining crystal violet was solved in 100 $\mu$L 33% acetic acid for 10 min. Next 70 $\mu$L solution was transferred into fresh wells. Optical density was measured at 600 nm. LDH leakage by Detroit 562 cells after treatment with 20 $\mu$M CPP-PNAs or 1 $\mu$g levofloxacin was determined using the CyQUANT LDH Cytotoxicity Assay kit (Thermo Fisher Scientific, Darmstadt, Germany) according to the manufacturer's instructions.

***G. mellonella* infection model.** Larvae of the greater wax moth (*G. mellonella*) were obtained from Bugs-International GmbH (Irsingen/Unterfeld, Germany). For infection experiments, *S. pneumoniae* TIGR4 was grown for 7 h in BHI, washed twice in a 0.9% NaCl solution, and suspended in 0.9% NaCl to a final concentration of approximately $1 \times 10^8$ CFU/mL. Larvae weighing 150 to 200 mg were infected with $1 \times 10^6$ to $3 \times 10^6$ CFU/larva. Bacteria were injected into the hemocoel of the larvae between the last two pairs of prolegs using a microapplicator (World Precision Instruments, LLC, Sarasota, FL) and a fine-dosage syringe (Omnican-F, 0.01 to 1 mL, 0.30 × 12 mm; B. Braun AG, Melsungen, Germany). As mock control, 0.9% NaCl (PNA experiment) or 0.9% NaCl/3% DMSO (levofloxacin experiment) was injected. For CPP-PNA treatment, larvae were injected 30 min postinfection with 10 nmol CPP-PNA/larva. For levofloxacin treatment, larvae were injected 30 min postinfection with 1 or 10 $\mu$g levofloxacin/larva. Larvae were incubated for 7 days, and survival was monitored daily (22).

For determination of bacterial load, larvae were infected and subsequently treated with NaCl (mock), CPP-PNAs, or levofloxacin (positive control). At 24 h after infection, larvae were homogenized. Following serial dilution, the suspension was plated on selective medium (BD Columbia CNA Agar with 5% sheep blood; Becton Dickinson GmbH, Heidelberg, Germany). Agar plates were incubated overnight at 37°C, 5% $CO_2$. CFU/larva was determined by visual inspection. Colony identity was verified on a random basis by colony PCR using *ply*-specific primers.

**Statistical analyses.** All experiments were performed at least three times or as indicated in the respective figure legends by sample size (*n*). Statistical significance was determined using the tests indicated in the respective figure legends. Statistical analyses were performed using GraphPad Prism 7 software.

## SUPPLEMENTAL MATERIAL

Supplemental material is available online only.

**SUPPLEMENTAL FILE 1**, PDF file, 0.4 MB.

## ACKNOWLEDGMENTS

This work was supported by the Federal Excellence Initiative of Mecklenburg Western Pomerania and the European Social Fund (ESF) (grant KoInfekt ESF/14-BM-A55-0010/16 to B.K. and grant KoInfekt ESF_14-BM-A55-0001_16 to S.H.) and by the University Medicine Rostock (grant Forun 889008 to N.P.).

G.B. contributed Investigation, Formal Analysis, Writing – Original Draft; C.A. contributed Investigation, Formal Analysis, Writing – Original Draft; I.P. contributed Investigation, Formal analysis, Writing – Review & Editing; A.B. contributed Investigation, Formal Analysis; S.H. contributed Methodology, Writing – Review & Editing; A.J. contributed Conceptualization, Methodology, Writing – Review & Editing; B.K. contributed Funding acquisition, Writing – Review & Editing; N.P. contributed Conceptualization, Methodology, Formal Analysis, Visualization, Supervision, Funding acquisition, Writing – Original Draft.

The authors have no conflicts of interest to declare.

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
