## [Reviewer comments · Microbiology Spectrum]

Microbiology Spectrum

Antimicrobial activity of peptide-coupled antisense Peptide Nucleic Acids in *Streptococcus pneumoniae*

Gina Barkowsky, Corina Abt, Irina Pöhner, Adam Bieda, Sven Hammserschmidt, Anette Jacob, Bernd Kreikemeyer, and Nadja Patenge

Corresponding Author(s): Nadja Patenge, ROSTOCK UNIVERSITY MEDICAL CENTER

Review Timeline:

Submission Date:	February 10, 2022
Editorial Decision:	March 3, 2022
Revision Received:	September 13, 2022
Editorial Decision:	October 12, 2022
Revision Received:	October 13, 2022
Accepted:	October 19, 2022

Editor: Amit Singh

Reviewer(s): Disclosure of reviewer identity is with reference to reviewer comments included in decision letter(s). The following individuals involved in review of your submission have agreed to reveal their identity: Mahavir Singh (Reviewer #2)

Transaction Report:

DOI: <https://doi.org/10.1128/spectrum.00497-22>

March 3, 2022

Dr. Nadja Patenge
ROSTOCK UNIVERSITY MEDICAL CENTER
Institute of Medical Microbiology, Virology and Hygiene
Schillingallee 70
Rostock 18057
Germany

Re: Spectrum00497-22 (Antimicrobial activity of peptide-coupled antisense Peptide Nucleic Acids in *Streptococcus pneumoniae*)

Dear Dr. Nadja Patenge:

Your manuscript is reviewed by two experts in the field. Both of them raised substantial concerns, which need your attention both experimentally and textually.

Link Not Available

Sincerely,

Amit Singh

Journals Department
Reviewer comments:

Reviewer #2 (Comments for the Author):

This paper entitled 'Antimicrobial activity of peptide-coupled antisense Peptide Nucleic Acids in *Streptococcus pneumoniae*' by Barkowsky et al. report cell-penetrating peptides (CPP-peptide) coupled to peptide nucleic acids (PNA) as a novel antibacterial agent in *S. pneumoniae*. The Paper is well written and easy to follow. I have the following major/minor criticism of the paper:

1. Authors have published a similar study targeting *Streptococcus pyogenes* using CPP-PNAs. Here authors have applied this

strategy to *Streptococcus pneumoniae*. As a proof of principle in just another Gram-positive *Streptococcus* species, this study is fine; however, I wonder if any further development has been done (for example, optimizing CPP-PNA conjugates) in this study compared to the previous study? Did these conjugates work better here in the *Streptococcus pneumoniae* infection model?

2. In the infection model (*Streptococcus pneumoniae* infected larvae of *G. mellonella*), the tested CPP-PNAs showed an enhancement of survival by ~ 20%. Is this significant? I am also wondering if the length of PNAs could be optimized for their better efficacy?

3. While I understand that authors have used CPP-scrambled PNA as a control conjugate in their assays, they also mention that CPPs have toxicity associated with it. Also, CPP-scr PNA showed an effect in the killing kinetic assays. Did the authors use only CPP as one of the control in growth assays?

4. Could the author report IC50 of the tested CPP-PNA from their toxicity assays?

5. Authors have used a low concentration (10 nmol) of CPP-PNAs in the *G. mellonella* infection model; however, in the bacterial toxicity assays, they have used concentrations in 2-20 μ M range? Authors can explain this.

6. Does the disease model organism *G. mellonella* have endogenous Gyr and rpoA genes? If yes, I wonder if the tested PNA sequences cross target them?

7. Any discussion or speculation as to why anti-apoB PNA is not affecting *Streptococcus pneumoniae* compared to anti-gyrA PNA?

Staff Comments:

Preparing Revision Guidelines

Please return the manuscript within 60 days; if you cannot complete the modification within this time period, please contact me. If you do not wish to modify the manuscript and prefer to submit it to another journal, please notify me of your decision immediately so that the manuscript may be formally withdrawn from consideration by Microbiology Spectrum.

Introduction:

Antimicrobial resistance and resistance to pneumococcal conjugate vaccines (PCV) pose a significant challenge in treatment of pneumonia caused by *Streptococcus pneumoniae*. Peptide nucleic acids (PNAs) can be a potential anti-microbial treatment, offering a species-specific advantage. The authors have investigated efficacy of different cell penetrating peptides (CPP) coupled with PNAs.

Merits:

The authors show that out of three CPP used two (TAT and (RXR)4XB) of them show promising antibacterial activity. It is also evident that *gyrA* is better antibacterial target than *rpoB*, for all three *S. pneumoniae* strains used in this study. Target gene specific downregulation has also been validated using RT-PCR. The authors have concluded study showing TAT and (RXR)4XB coupled *gyrA* conferring survival advantage using *G. mellonella* infection model.

Critique:

The manuscript investigates antimicrobial properties of CPP-PNAs against *S. pneumoniae*. The antimicrobial effects seen here are in assays which involve incubation of pathogen directly with CPP-PNA's for long hours. To have a comprehensive assessment of efficacy, it is important to see antimicrobial effects in cell-based infection system. It is also required to see cell toxicity of these compounds at concentrations showing bactericidal activity(10-20µm) .

G. mellonella infection model shows around 10-20% increase in survival. Due to lack of CFU data and an antibiotic control in this experiment, it is difficult to ascertain whether this 20% increase in survival is clinically relevant. It is therefore recommended either to repeat this experiment with CFU data and appropriate antibiotic control or authors could check efficacy in widely accepted murine model.

The manuscript needs writing corrections to be reader friendly. Few examples are as listed below:

- 1) Line 68: "WHO" should be "WHO's"
- 2) Line 223: It should be mRNA levels
- 3) Line 305: "as targeting" should be replace by "than targeting"
- 4) Line 502: "Anti" should be "anti"

Reviewer 1

We are grateful for the reviewer's positive feedback. We agree that the suggested experiments will improve the manuscript. In the following point to point response, we describe shortly the setup of the assays and provide the corresponding paragraphs that have been added to the text. We also corrected the text as suggested by the reviewer and additionally re-corrected the manuscript thoroughly to remove writing errors.

>The manuscript investigates antimicrobial properties of CPP-PNAs against *S. pneumoniae*. The antimicrobial effects seen here are in assays which involve incubation of pathogen directly with CPP-PNA's for long hours. To have a comprehensive assessment of efficacy, it is important to see antimicrobial effects in cell-based infection system. It is also required to see cell toxicity of these compounds at concentrations showing bactericidal activity (10-20µm) .

Answer: To test antimicrobial efficacy in a cell-based infection system, we infected Detroit 562 cells with *S. pneumoniae* TIGR4 at a multiplicity of infection (MOI) of 5. Cells were treated with 20 µM (RXR)₄XB-anti-*gyrA* PNA for 3 h. CFU of bacteria were determined for CPP-antisense PNA-treated samples and compared to CFU of non-treated samples and scrambled control PNA samples. We summarized these data in Figure 4 A and described the experiment in the text:

Materials and Methods:

Pages 7-8, lines 147-156: "The human pharynx carcinoma, epithelial cell line Detroit 562 (ATCC CCL138) was cultivated in Dulbecco's modified eagle medium (DMEM)/F12 (1:1) (Thermo Fisher Scientific, Schwerte, Germany) supplemented with 10% FBS at 37°C and 5% CO₂. Confluent Detroit 562 cells were infected with *S. pneumoniae* TIGR4 at a multiplicity of infection (MOI) of 5. Cultures were treated with 20 µM CPP-PNAs or 1 µg levofloxacin, respectively, and incubated for 3 h at 37°C, 5% CO₂. After incubation, the supernatant was collected. Adherent cells were detached with Accutase™ solution (Merck KGaA, Darmstadt, Germany), collected by centrifugation for 10 min at 4 000g and combined with the supernatant. Cells were pelleted by centrifugation, washed twice with PBS and suspended in PBS. CFU/well were determined following serial dilution and plating on THY agar plates."

Results:

Pages 13-14, lines 287-295: "Detroit 562 human pharyngeal epithelial cells were infected with *S. pneumoniae* TIGR4 at a multiplicity of infection (MOI) of 5. Cells were treated with 20 µM (RXR)₄XB-anti-*gyrA* PNA or scrambled control PNA, respectively. As positive control, 1 µg levofloxacin, corresponding to 3 x MIC, was added. Following incubation for 3 h at 37°C, bacterial CFU were determined for CPP-antisense PNA-treated samples and compared to CFU in non-treated samples and in samples containing scrambled control PNAs (Figure 4A). Treatment with (RXR)₄XB-anti-*gyrA* caused a log CFU reduction of 1. Addition of levofloxacin led to a log CFU reduction of 2. No significant changes of bacterial CFU were observed following application of (RXR)₄XB-conjugated scPNAs in comparison to the untreated control."

Cytotoxicity was investigated using a crystal violet assay and by measuring LDH release. We summarized these data in Figure 4 B and added the following information to the text:

Materials and Methods:

Page 8, lines 158-166: "Confluent Detroit 562 cells were treated with 20 µM CPP-PNAs or 1 µg levofloxacin, respectively, and incubated for 3 h at 37°C, 5% CO₂. Crystal violet staining was performed to determine biomass. Cells were washed with PBS and fixed with 100 µl Methanol for 10 min. Subsequently, cells were washed with phosphate buffered saline, containing 0.005% Tween 20,

incubated with 0.1% crystal violet solution for 10 minutes, and washed with H₂O. The remaining crystal violet was solved in 100 µl 33% acetic acid for 10 min. 70 µl solution were transferred into fresh wells. Optical density was measured at 600 nm. LDH leakage by Detroit 562 cells after treatment with 20 µM CPP-PNAs or 1 µg levofloxacin was determined employing the CyQUANT™ LDH Cytotoxicity Assay Kit (Thermo Fisher Scientific, Darmstadt, Germany) according to the manufacturer's instructions."

Results:

Page 14, lines 296-305: "Potential cytotoxic effects of CPP-PNAs on Detroit 562 cells were investigated by measuring cellular viability using a crystal violet assay (Figure 4B). Cells were incubated with 20 µM (RXR)₄XB-PNA or TAT-PNA, respectively. Untreated cells or treatment with levofloxacin served as negative control. Lysis of the cells was used as positive control for cell death. No cell death was observed following treatment with levofloxacin or CPP-PNAs. Similar results were observed using a LDH leakage assay. LDH release after treatment of Detroit 562 cells was compared to maximum LDH release. Application of 20 µM (RXR)₄XB-PNA, 20 µM TAT-PNA, or 1 µg levofloxacin caused 5%, 6%, or 3%, respectively, of maximum LDH release. Together, these results indicate that CPP-PNA conjugates have low cytotoxic activity comparable to that of the conventional antibiotic levofloxacin."

>*G. mellonella* infection model shows around 10-20% increase in survival. Due to lack of CFU data and an antibiotic control in this experiment, it is difficult to ascertain whether this 20% increase in survival is clinically relevant. It is therefore recommended either to repeat this experiment with CFU data and appropriate antibiotic control or authors could check efficacy in widely accepted murine model."

Answer: Larvae were infected with *S. pneumoniae* TIGR4 and subsequently treated with levofloxacin. We included the corresponding survival curves (Figure 5A). We also added CFU data (Figure 6) to compare PNA treatment with conventional antibiotic application. The experiments are described in the text:

Materials and Methods:

Page 9, lines 177/178: "For levofloxacin treatment, larvae were injected 30 min post-infection with 1 or 10 µg levofloxacin/larva."

Page 9: lines 179-185: "For determination of the bacterial load, larvae were infected and subsequently treated with NaCl (mock), CPP-PNAs, or levofloxacin (positive control). 24 h after infection, larvae were homogenized. Following serial dilution, the suspension was plated on selective medium (BD™ Columbia CNA Agar with 5% Sheep Blood, Becton Dickinson GmbH, Heidelberg, Germany). Agar plates were incubated over night at 37°C, 5% CO₂). CFU/larva were determined by visual inspection. Colony identity was verified on a random basis by colony PCR using ply-specific primers."

Results:

Pages 14-15, lines 307-316: "Antimicrobial efficiency of CPP-antisense PNA conjugates was evaluated *in vivo* employing a *G. mellonella* infection and treatment model[22, 30]. Larvae were infected with *S. pneumoniae* strain TIGR4 and subsequently treated with 10 nmol CPP-anti *gyrA* PNAs. As positive control served treatment with levofloxacin. Survival of larvae was observed over 7 days.

Administration of 10 µg and 1 µg levofloxacin, corresponding to approximately 25x or 2.5x MIC, respectively, increased survival of infected larvae from 19% to 74% and 40%, respectively (Figure 5A).

Larvae treated with TAT-anti-gyrA PNA or (RXR)4XB-anti-gyrA PNA exhibited increased survival in comparison to mock-treated larvae or to larvae treated with CPP-scPNA (Figure 5B/C). The survival rate was 25% or 22% higher following injection of TAT-anti-gyrA PNA or (RXR)4XB-anti-gyrA PNA, respectively, which is comparable to the effect of 1 µg levofloxacin (21%)”

Page 15: 321-329: “To establish the antibacterial effect of PNAs in the invertebrate infection model, the bacterial load of infected larvae was assessed (Figure 6). Larvae were infected and subsequently treated with NaCl (mock), 10 nmol CPP-PNA or levofloxacin (positive control), respectively. From each group, four larvae were homogenized at 24 h post-infection. CFU/larva were determined following serial dilution of the suspension and plating on selective medium. Treatment with (RXR)4XB-anti-gyrA caused a log CFU reduction of 1 compared to the mock-treated larvae. Addition of 1 µg or 10 µg levofloxacin led to a log CFU reduction of 2 or 6, respectively. In contrast, treatment with (RXR)4XB-conjugated scPNAs caused no significant changes of bacterial CFU in comparison to the untreated control, suggesting a target gene specific antimicrobial effect in the infected larvae.”

> The manuscript needs writing corrections to be reader friendly. Few examples are as listed below

Answer: we changed the text accordingly

Page 4, line 68: WHO’s

Page 12, line 265: mRNA levels

Page 17, line 373: than targeting

Page 19, line 414: utilized

Page 28, line 597: anti

Reviewer 2

1. Authors have published a similar study targeting *Streptococcus pyogenes* using CPP-PNAs. Here authors have applied this strategy to *Streptococcus pneumoniae*. As a proof of principle in just another Gram-positive *Streptococcus* species, this study is fine; however, I wonder if any further development has been done (for example, optimizing CPP-PNA conjugates) in this study compared to the previous study? Did these conjugates work better here in the *Streptococcus pneumoniae* infection model?

Answer: In this study, we asked whether CPP-PNA treatment of *S. pneumoniae* is principally feasible. Therefore, we tested basic constructs that have been widely used for Gram-negative and Gram-positive organisms. CPPs are coupled via an ethylene glycol linker to a 10 mer antisense PNA (compare to the answer of question 2) that covers the ATG region of the *gyrA* gene. We concentrated on CPPs that worked efficiently in *S. pyogenes*, speculating that the cell surface of Gram-positive organisms supports uptake of similar compounds. Overall, we found that *S. pneumoniae* is susceptible to antisense PNA treatment but that the constructs used in this work require optimization. In the discussion, we already proposed strategies for carrier-PNA development, including more potential carrier molecules and different target genes. We now highlight in the discussion, citing a recent publication by Popella et. al, the influence of target genes on the efficacy of antisense PNAs. This group could also show that 9mer PNAs are sufficiently effective (compare to question 2).

Discussion:

Pages 18-19, lines 404-409: "In uropathogenic *Escherichia coli*, 11 essential genes with varying expression levels were investigated as potential targets for peptide-conjugated PNAs (40). Three promising target mRNAs were identified for effective growth inhibition, i.e. *dnaB*, *ftsZ* and *rpsH*. The analysis also showed that transcript abundance does not predict target vulnerability. Furthermore, the authors demonstrated, that 9mer PNAs were generally as effective in inhibiting bacterial growth as their 10mer counterparts (40)."

2. In the infection model (*Streptococcus pneumoniae* infected larvae of *G. mellonella*), the tested CPP-PNAs showed an enhancement of survival by ~ 20%. Is this significant? I am also wondering if the length of PNAs could be optimized for their better efficacy?

Answer: Statistical significance of the differences has been determined using the Log-rank test (compare to Figure legend 5). We now also included a positive control (levofloxacin) and CFU data to substantiate the antibacterial effect of antisense PNAs in the *G. mellonella* experiments (Figure 5A and Figure 6).

It has been shown that the optimal lengths of antisense PNAs is 10-14. Thereby, translocation efficiency decreases with lengths, whereas specificity increases with lengths. Recently, Popella et al. showed that a 9-mer was effective in *E. coli*. We believe that varying target genes and PNA length could improve antibacterial efficacy of the constructs (compare to the answer to question 1).

3. While I understand that authors have used CPP-scrambled PNA as a control conjugate in their assays, they also mention that CPPs have toxicity associated with it. Also, CPP-scr PNA showed an effect in the killing kinetic assays. Did the authors use only CPP as one of the control in growth assays?

Answer: We decided to use CPP-scrambled PNAs as control agent because of the chemical similarity to the antisense constructs. It had been shown before, that the cytotoxicity of CPPs is highly dependent on the cargo used (El-Andaloussi et al. 2007). In accordance with this observation, antibacterial effects of (RXR)₄XB and TAT peptides on *S. pneumoniae* were higher than those of the coupled peptides. We included these data in Supplementary Material Figure S1 and added the following paragraph to the text:

Pages 11-12, lines 243-246: "CPP-coupled scrambled PNAs were used as control throughout this study, because the cytotoxicity of CPPs is highly dependent on the cargo used (30). Accordingly, antibacterial effects of (RXR)₄XB and TAT peptides on *S. pneumoniae* TIGR4 were higher than those of the conjugated peptides (Suppl. Figure 3)."

4. Could the author report IC50 of the tested CPP-PNA from their toxicity assays?

Answer: We are grateful to the reviewer's suggestion and included the information in the manuscript.

Materials and Methods:

Pages 5-6, lines 107-110: "The concentration of CPP-PNA that reduced CFU/ml by 50% after 6 h incubation compared to an untreated control (50% inhibitory concentration [IC₅₀]) was calculated by nonlinear regression analysis using GraphPad Prism software (GraphPad Software, San Diego, CA)."

Results:

Page 10, line 213: [IC₅₀] of 0.4 μM was determined for TAT-anti-*gyrA* PNA.

Pages 10-11, lines 221-222: [IC₅₀] of (RXR)₄XB-anti *gyrA* PNA was 0.63 μM.

5. Authors have used a low concentration (10 nmol) of CPP-PNAs in the *G. mellonella* infection model; however, in the bacterial toxicity assays, they have used concentrations in 2-20 μM range? Authors can explain this.

Answer: We aimed to use approximately the same doses of PNAs in all assays. The range *in vitro* was 2-20 μM, corresponding to 2-20 nmol/ml. In the *G. mellonella* infection model, we decided to use one representative concentration. It is difficult to obtain a defined concentration within larvae because PNAs are not evenly distributed throughout the animal. To account for this fact, we assumed a volume of about 250 μl/larva and injected 10 nmol/larva, corresponding to 40 nmol/ml, a relatively high approximate concentration compared to the *in vitro* experiments.

6. Does the disease model organism *G. mellonella* have endogenous *Gyr* and *rpoA* genes? If yes, I wonder if the tested PNA sequences cross target them?

Answer: *G. mellonella* does not have endogenous *gyrA* and *rpoB* genes. Gyrase A is a subunit of the essential enzyme gyrase that belongs to the class of topoisomerases and is present in both bacteria and archaea. RpoB is a subunit of the bacterial DNA-dependent RNA polymerase. Both genes, *gyrA* and *rpoB*, are suitable target genes, because there are no eukaryotic counterparts. Moreover, the sequence across the start codon is species-specific. Thus, targeting *gyrA* in *S. pyogenes* and in *S. pneumoniae* requires distinct PNA sequences.

7. Any discussion or speculation as to why anti-*apoB* PNA is not affecting *Streptococcus pneumoniae* compared to anti-*gyrA* PNA?

Answer: It has been widely observed that different antisense targets have different effects in bacteria. One recent study systematically analyzed the effects of 11 target genes following PNA-treatment in *E. coli* using RNAseq (Popella et al. 2022). The reasons seem to be multifactorial, and researchers are currently trying to determine the optimal target genes through trial and error. We cite the work in the Discussion (see the answer to questions 1 and 2).

October 12, 2022

Dr. Nadja Patenge
ROSTOCK UNIVERSITY MEDICAL CENTER
Institute of Medical Microbiology, Virology and Hygiene
Schillingallee 70
Rostock 18057
Germany

Re: Spectrum00497-22R1 (Antimicrobial activity of peptide-coupled antisense Peptide Nucleic Acids in *Streptococcus pneumoniae*)

Dear Dr. Nadja Patenge:

Please address a minor issue raised by the reviewer # 1. My suggestion would be to add the data as a supplementary figure.

Thank you for submitting your manuscript to Microbiology Spectrum. As you will see your paper is very close to acceptance. Please modify the manuscript along the lines I have recommended. As these revisions are quite minor, I expect that you should be able to turn in the revised paper in less than 30 days, if not sooner. If your manuscript was reviewed, you will find the reviewers' comments below.

When submitting the revised version of your paper, please provide (1) point-by-point responses to the issues raised by the reviewers as file type "Response to Reviewers," not in your cover letter, and (2) a PDF file that indicates the changes from the original submission (by highlighting or underlining the changes) as file type "Marked Up Manuscript - For Review Only". Please use this link to submit your revised manuscript. Detailed instructions on submitting your revised paper are below.

Link Not Available

Sincerely,

Amit Singh

Reviewer comments:

Reviewer #1 (Comments for the Author):

Comments:

The revised manuscript from Barkowsky et al. have done cell-based assays with antibiotic control which gives a better assessment of efficacy of CPP-PNAs as potential as new antimicrobial strategy targeting *S. pneumoniae*. Also, using same cells authors have addressed cell toxicity concerns with CPP-PNAs, I do have one suggestion, the authors have mentioned about results of LDH leakage assay on page-14: line-301-305, however, data is not shown. I would recommend either mentioning "data not shown or add data in supplementary".

Reviewer #2 (Comments for the Author):

I am happy with the answers to the questions that I raised in my review. The authors have made appropriate changes in the revised manuscript.

Preparing Revision Guidelines

Please return the manuscript within 60 days; if you cannot complete the modification within this time period, please contact me. If you do not wish to modify the manuscript and prefer to submit it to another journal, please notify me of your decision immediately so that the manuscript may be formally withdrawn from consideration by Microbiology Spectrum.

Reviewer 1

We would like to thank you again for your helpful suggestions to improve our manuscript.

> I do have one suggestion, the authors have mentioned about results of LDH leakage assay on page-14: line-301-305, however, data is not shown. I would recommend either mentioning "data not shown or add data in supplementary".

Answer: We show the data in the supplementary material and changed the text accordingly:

Page 14, lines 302-303 : "LDH release after treatment of Detroit 562 cells was compared to maximum LDH release (Suppl. Figure 4)."

Reviewer 2

Thank you very much for your help.

October 19, 2022

Dr. Nadja Patenge
ROSTOCK UNIVERSITY MEDICAL CENTER
Institute of Medical Microbiology, Virology and Hygiene
Schillingallee 70
Rostock 18057
Germany

Re: Spectrum00497-22R2 (Antimicrobial activity of peptide-coupled antisense Peptide Nucleic Acids in *Streptococcus pneumoniae*)

Dear Dr. Nadja Patenge:

Your manuscript has been accepted, and I am forwarding it to the ASM Journals Department for publication. You will be notified when your proofs are ready to be viewed.

Sincerely,

Amit Singh
Editor, Microbiology Spectrum

Journals Department
Supplemental Material 1-4: Accept